# Individual lipid alterations at the origin of neuronal Ceramide Synthase defects

Anna B. Ziegler[1,2]*, Cedrik Wesselmann[2], Konstantin Beckschäfer[3], Anna-Lena Wulf[4¤a], Neena Dhiman[5,6], Peter Soba[5,6], Christoph Thiele[3], Reinhard Bauer[4], Gaia Tavosanis[1,7¤b]*

1 Dendrite Differentiation, German Center for Neurodegenerative Diseases, Bonn, Germany, 2 Institute of Neuro- and Behavioral Biology, University of Münster, Münster, Germany, 3 Biochemistry & Cell Biology of Lipids, LIMES-Institute, University of Bonn, Bonn, Germany, 4 Molecular Developmental Biology, LIMES-Institute, University of Bonn, Bonn, Germany, 5 Institute of Physiology and Pathophysiology, Friedrich-Alexander-Universität Erlangen-Nürnberg, Erlangen, Germany, 6 Department of Molecular Brain Physiology and Behavior, LIMES Institute, University of Bonn, Bonn, Germany, 7 Developmental Neurobiology, LIMES-Institute, University of Bonn, Bonn, Germany

¤a Current address: Department of Developmental Pathology, Institute of Pathology, University Hospital Bonn, Bonn, Germany
¤b Current address: Department of Developmental Biology, Institute for Biology II, RWTH Aachen University, Aachen, Germany
* anna.ziegler@uni-muenster.de (ABZ); gaia@devbiol.rwth-aachen.de (GT)

## Abstract

The brain is highly susceptible to disturbances in lipid metabolism. Among the rare, genetically-linked epilepsies Progressive Myoclonic Epilepsy Type 8 (PME8), associated with the loss of Ceramide Synthase (CerS) activity, causes epileptic symptoms accompanied by early onset of neurodegenerative traits. The function of CerS is embedded in a complex, conserved metabolic pathway, making it difficult to identify the specific disease-relevant alterations. Here, we show that the expression of an enzymatically inactive *cerS* allele in *Drosophila* sensory neurons yielded developmental and early onset dendrite loss. Combining lipidomics and refined genetics with quantitative analysis of neuronal morphology in *cerS* mutants, we identified which lipids species are dysregulated and how they affect neuronal morphology. In *cerS* mutants, long and very-long acyl-chain C18-C24-ceramides were missing and necessary for dendrite elaboration. In addition, the substrate of CerS, (dh)S, and its metabolite (dh)S1P, increased. Especially increasing (dh)S1P strongly reduces dendritic complexity in *cerS* mutant neurons. Finally, we performed *in vivo* experiments to cell-autonomously rescue the morphological defects of *cerS* mutant neurons and report that a complete rescue can only be achieved if the toxic CerS substrate is converted to produce specific (C18-C24) ceramides. Thus, despite the complex metabolic alterations, our data provides essential information about the metabolic origin of PME8 and delineates a potential therapeutic avenue.

**Data availability statement:** All relevant data are in the manuscript and its Supporting information files.

**Funding:** This study obtained funding from the Deutsche Forschungsgemeinschaft DFG ZI1690/2-1 to ABZ and from the Nordrhein-Westfalen (NRW) network iBehave to GT. The funders had no role in study design, data collection and analysis, decision to publish, or preparation of the manuscript.

**Competing interests:** The authors have declared that no competing interests exist.

## Authors summary

The brain is the most lipid-enriched organ of our body. At the same time, it is highly susceptible to disturbances in lipid metabolism. In this context, ceramide metabolism deserves special attention since mutations affecting ceramide anabolic and catabolic processes have very severe neurological consequences. Ceramides are the precursors of complex Sphingolipids (SL). Ceramide Synthases (CerS) are at the center of ceramide *de novo* formation and produce ceramides with particular acyl-chain lengths. Mutations in *cerS* genes result in very strong epileptic seizures and developmental neurodegeneration. Using *Drosophila*, we show that abolishing CerS enzymatic activity does more than merely reduce complex SL. Additionally, the substrate of the CerS reaction, (dh)Sphingosine, and its toxic metabolites are elevated, and we show that both of these alterations are deleterious for neurons. To gain indications towards a potential therapy, multiple rescue experiments were performed. However, an efficient rescue of neuronal morphology could only be obtained when specifically CerSs were re-expressed in mutant neurons that could turn (dh)Sphingosine into ceramides with the right acyl-chain length thus restoring the original lipid composition.

## Introduction

60% of the total dry mass of the human brain consists of lipids [1]. Lipids are involved in a variety of biological processes, utilized as an energy substrate, building blocks for membranes, and function as bioactive signaling molecules [2]. Because of the various and sensitive roles of individual lipids, neurons are highly vulnerable to alterations in their homeostasis. Ceramide metabolism deserves specific attention in this context. Disruption of ceramide synthesis or degradation leads to a number of serious diseases with a low prevalence, such as Progressive Myoclonic Epilepsy Type 8 (PME8), Hereditary sensory neuropathy (HSN1), Niemann-Pick Disease, Fragile X-associated tremor/ataxia syndrome (FXTAS), or Gaucher Disease [3,4]. Each of these diseases is associated with an imbalance in ceramide levels, leading primarily to neurological symptoms. This emphasizes the importance of a well-controlled ceramide metabolism in the nervous system [3,5]. In terms of their structure, ceramides are the simplest members of the large sphingolipid (SL) family. Ceramide Synthases (CerSs) are at the center of ceramide synthesis and, by extension, of complex SL production [6]. Loss of CerS activity leads to PME8, a disease with an early onset (between 1 and 16 years of age) in which patients develop myoclonic seizures [7–9]. In addition, PME8 patients suffer from neurodegeneration leading to moderate to severe and progressive cognitive impairment, ataxia, and gait disturbances [7,9]. Interestingly, mutations that affect the *de novo* ceramide synthesis pathway upstream of CerS are also associated with polyneuropathic phenotypes such as paresthesia, shooting pain, and disturbed pain delusions in the extremities [10,11]. Whole exon sequencing and homozygosity mapping in patients diagnosed with PME8 identified two point mutations within *cerS1* (*cerS1*[H183Q] and *cerS1*[R255C]) [7–9,12]. Additionally, a

deletion on chromosome 1q21, which removes the entire *cerS2* gene also results in PME [8]. However, the broad expression pattern of *cerS* genes complicates the elucidation of their nervous system specific function [13–16].

Ceramide *de novo* synthesis generates a large fraction of cellular ceramides. The pathway entails four enzymatic reactions by which at first (dh)sphingosine ((dh)S) is produced and then acylated to (dh)ceramide through the sphingosine N-acyltransferase activity of CerS [5]. The catalytic center of CerS lays within the lag1p motif and includes two adjacent histidines that are required for enzymatic activity; one of these is mutated in the PME, producing the human *cerS1*[H183Q] allele (Fig 1A) [7,17]. Mammalian genomes harbor six CerS encoding genes (*cerS1-6*), which differ in their expression pattern and substrate specificity and use only acyl-CoA species with a defined chain length for (dh)ceramide production [18,19]. The region that defines the acyl-chain specificity maps to the loop between transmembrane domains (TMDs) six and seven, a region to which the second human *cerS1* mutation (*cerS1*[R255C]) has been mapped (Fig 1A) [18].

Ceramides can be directly incorporated into membranes, or they can be used as substrate to produce complex SL. While the most prominent SLs in vertebrates are sphingomyelins (SM), invertebrates have high levels of ceramide-phosphatidylethanolamine (Cer-PE) [20]. Ceramides with a specific acyl-chain length can also directly interact with ceramide-binding proteins and thereby fulfill various roles as bioactive compounds [21,22]. Importantly, not only are the ceramides themselves lipid messenger molecules, but also the substrate for the CerS reaction, (dh)S, and its related metabolites (dh)S1P, Sphingosine (S), and Sphingosine-1-P (S1P) function in this capacity. Their role is largely linked to growth regulation and control of survival versus apoptosis [23]. In the mouse brain, all of these so-called long-chain bases (LCBs) accumulate upon CerS1 dysfunction [16]. In detail, two mouse mutant lines (*toppler* (*to*) and *flincher* (*fln*)) were isolated that harbor mutations in the acyl-chain binding loop of CerS1, resulting in viable but small adults suffering from ataxia [15,16]. In these mutants, dendrite degeneration of the cerebellar Purkinje cells in the central nervous system (CNS) was observed and a causal link between general LCBs accumulation and dendritic degeneration established. However, the lack of complex SL might also contribute to neurodegeneration in *cerS* mutant neurons [15,16]. As is typical for lipid metabolism-related diseases, it remains unclear which of the complex alterations in lipid balance associated with loss of CerS function promotes neurite loss and degeneration and how PME8 could be potentially treated.

In contrast to mammals, the genome of invertebrates such as *Drosophila melanogaster* encodes for only one CerS, which is proposed to have broad substrate specificity [24]. Reduction of CerS function enhances Fragile X-associated tremor/ataxia syndrome-linked neuronal degeneration in the fly eye [4]. Furthermore, knockdown of *cerS* in *Drosophila* ventral lateral neurons (LNv's), located in the larval CNS and in neurons forming the Mushroom bodies of the adult CNS, reduces the amount of neurites and leads to targeting defects [25,26] suggesting a conserved role of CerS in the establishment or maintenance of neuronal morphology. These studies, together with the structural and functional conservation of this enzyme, suggest that *Drosophila* could be a useful model to understand the origin of the dendrite loss and neurodegeneration associated with *cerS* mutations.

Here, we used *Drosophila* class four dendritic arborization (c4da) neurons to study the effects of CerS dysfunction on neurons. The complex dendrites of the c4da neurons are housed directly under the transparent cuticle of the larva which facilitates *in vivo* imaging [27]. They can be visualized and genetically manipulated with highly specific Gal4 driver lines, allowing a detailed quantitative analysis of their dendritic morphology in immobilized, fully intact and living animals. Due to these properties c4da neurons are an ideal model system to answer fundamental neurobiological questions and to study the specific impact of disease-related genes on neuronal development, function and maintenance [28–34]. At the same time, they are the larval primary nociceptors [30]. Thus, they functionally resemble neurons whose function is impaired in humans suffering from HSAN1. HSAN1 is caused by mutations upstream of CerS which also affect the ceramide de novo synthesis pathway [10,30].

In this study we show that the levels of multiple lipid species were changed in *Drosophila cerS* mutants and demonstrate that *Drosophila* CerS is required for the synthesis of long-chain ceramides. With quantitative genetic analysis, we furthermore demonstrate the impact of individual lipid alterations on neuronal structural complexity. Finally, we identify the

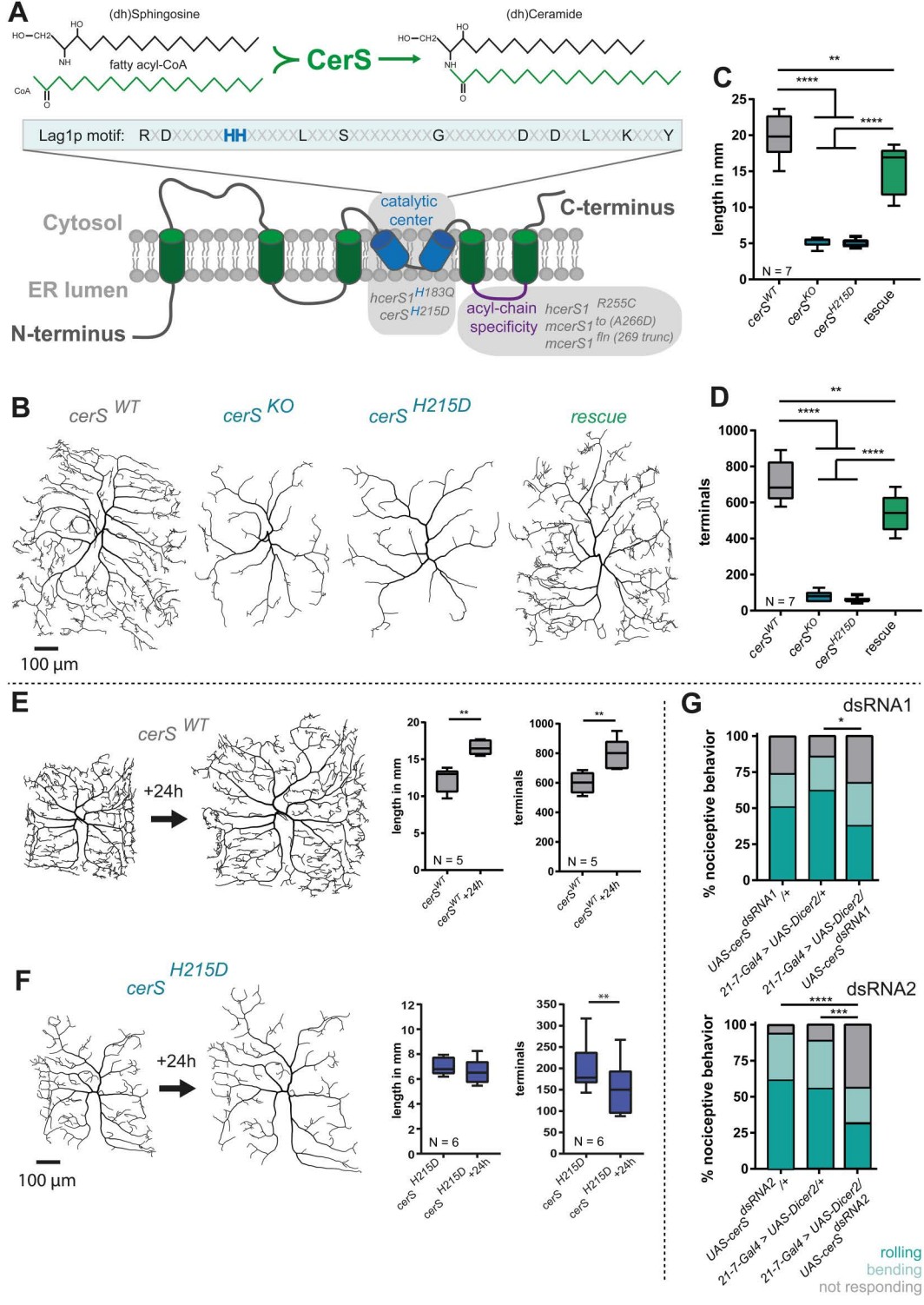

**Fig 1. Cell-autonomous CerS activity is required for dendrite morphology. A,** Up, CerS catalyzes the formation of (dh)ceramide from (dh)sphin-gosine and fatty acyl-CoA. Down, schematic representation of CerS topology (modified after Thidar et al., 2018). The *hcerS1H183Q* and the *Drosophila cerSH215D* mutations localize within the lag1p-motif, which is required for CerS catalytic activity (blue). *hcerS1R255C* and *mcerS1to* mutations lay within a loop that determines acyl-chain specificity (purple). **B-D,** Mutant c4da single cell clones expressing wild-type CerS (*cerSWT*), no CerS (*cerSKO*) or the catalytic dead *cerSH215D* were generated using the MARCM technique. *CerS* expression was cell-autonomously rescued in *cerSKO* mutant neurons using

a *UAS-cerS* expression construct. **B** Representative tracings of the single-cell MARCM clones. **C** and **D,** Quantification of total dendrite length and total number of dendritic terminal endpoints, respectively. **E and F**, *CerS^WT* and *cerS^H215D* c4da MARCM clones were imaged in early LIIIf stage larvae. Afterwards, larvae were allowed to develop for 24h and the same neuron was re-imaged. Images show representative dendritic trees and graphs show the quantified total dendrite length and total number of dendritic terminal endpoints, respectively. **G,** Nociceptive responses (rolling and bending) of animals, in which *cerS* was knocked down using RNAi were assayed using a mechanonociceptive assay with a 50 mN von Frey filament. Statistical significance is indicated as follows in this and all following figures: $* = p \leq 0.05$; $** = p \leq 0.01$; $*** = p \leq 0.001$; $**** = p \leq 0.0001$. Statistical test used in C and D: 1-Way-ANOVA followed by Tukey's multiple comparison test; Statistical test used in E and F: paired *t*-test. Statistical test used in G: $Chi^2$-test.

parallel loss of specific long- and very long-chain ceramides and the accumulation of specific LCBs as key factors hampering neuronal integrity, thus pointing to possible treatments.

## Results

### Cell-autonomous CerS activity is required for dendrite maintenance

To elucidate the impact of CerS dysfunction in *Drosophila* neurons, we first generated a new *cerS* allele in which the endogenous *cerS* gene was replaced by a mutant variant. To do so we used a previously published *cerS^KO* allele in which most of the protein coding exons were removed and replaced by an attP landing site [35]. We then used this landing site to re-integrate the missing exons encoding the wild-type sequence (*cerS^WT*) as a control [35], or a sequence bearing a specific mutation. The mutation we introduced replaces histidine at position 215 with aspartic acid. Histidine^215 is located within the lag1p motif and thus resembles the human *cerS1^H183Q* mutation. Additionally, a mutation in histidine to aspartic acid at this position has previously been shown to abolish enzymatic activity (Fig 1A) [7,17,35]. *cerS^KO* and *cerS^H215D* animals, as well as the previously characterized strong hypomorphic allele *cerS^G0349*, die during early larval stages, which morphologically correspond to the first larval instar [24]. Since the *cerS^H215D* mutation caused early larval lethality, we could not directly test its impact on CerS enzymatic activity in whole body mutants. However, using a fat body-specific Gal4-driver we have previously shown that early developmental lethality of *cerS^G0349* could be restored by expressing *UAS-cerS* but not UAS-*cerS^H215D*. Also, overexpression of *UAS-cerS^H215D* in wild-type larvae did not increase ceramide levels while overexpression of *UAS-cerS* did [24], supporting the idea that *cerS^H215D* is enzymatically inactive.

In mice and flies loss of CerS function leads to a reduction of neuronal dendrite complexity [16,25]. We thus sought to quantitatively characterize the impact of the newly generated *cerS^H215D* allele on dendrite complexity using the multi-modal nociceptive c4da neurons of the fly larva [36]. To overcome the previously mentioned lethality we used the MARCM (Mosaic Analysis with a Repressible Cell Marker) technique by which single homozygous mutant cell clones are generated in an otherwise heterozygous animal [37,38]. Single cell clones express UAS-mCD8GFP and can therefore be easily identified. Using MARCM, we obtained homozygous *cerS^KO* or homozygous *cerS^H215D* mutant c4da single neuronal cell clones. In comparison to *cerS^WT* c4da neurons, *cerS^KO* and *cerS^H215D* mutant neurons exhibited approximately a 75% reduction of total dendritic length and almost 90% of their total dendritic terminal endpoints (Fig 1B–1D). Importantly, a heterozygote mutant background did not affect the dendritic morphology of c4da neurons (S1A Fig). To test if only complex neuronal dendrites were affected by loss of CerS function, we analyzed the simpler dendrites of c1da and c3da *cerS* mutant neurons. Similarly, c1da and c3da neurons displayed a clear reduction in the number of dendrites (S1A and S1B Fig), suggesting that CerS function may generally be necessary for neurons to properly build or maintain their dendrites. We next confirmed that the observed phenotype was due to the cell-autonomous loss of CerS activity in the c4da neurons. We therefore re-expressed wild-type CerS in the *cerS^KO* mutant single cell MARCM clones, which rescued the total dendrite length and the total number of dendrite terminal endpoints. These experiments demonstrate that CerS activity is cell-autonomously required in c4da neurons (Figs 1B–1D, S1A and S1B).

To distinguish between neurodevelopmental growth defects and neurite loss in the *cerS* mutants, we compared the dendrites of mutants and controls at several developmental stages. Mutant c4da neurons already showed significant

dendrite growth deficits at the second instar larval stage which strongly suggests an early developmental component (S1C Fig). In addition, the difference in dendritic tree morphology between mutants and control became more pronounced during the third larval instar (LIII), which is divided into a feeding period (LIIIf) and the later wandering stage (LIIIw) (S1D Fig). *cerS^WT* c4da dendrites constantly grew in length and increased in dendritic complexity, whereas *cerS^KO* mutant c4da neurons showed an even simpler dendritic tree morphology at the LIIIw stage than at the earlier LIIIf stage. These data suggest that here, in addition to early developmental defects, degenerative processes might be at play. To specifically address the loss of dendrite branchlets during the LIII stage, we imaged *cerS* deficient MARCM clones at the early LIIIf stage and re-imaged the same cells 24h later. This analysis confirmed the loss of identifiable dendrites in *cerS^H215D* mutant c4da neurons (Fig 1E and 1F) or in *cerS^KO* mutant c4da neurons (S1E Fig). Caspases are important regulators of apoptosis and caspase activation was detected in *cerS* mutant cultured hippocampal neurons [16]. We therefore tested whether c4da neurons undergo caspase mediated apoptosis by using Apoliner, a caspase activity reporter [39]. However, no caspase activity was detected (S2A Fig). Furthermore, the expression of the caspase inhibitor p35 in mutant c4da neurons did not rescue the dendritic defects in *cerS* mutant c4da neurons [40] (S2B and S2C Fig). We next tested if CerS is required for the maintenance of c4da neurons and followed five individual *cerS^KO* c4da mutant MARCM clones from the third larval instar to the late pupal stage. Although the dendrite morphology of the dorsal *cerS^KO* mutant MARCM clones was largely affected (S2D and S2E Fig) all clones could be detected in larva and late pupa, indicating that these cells do not die during this period. To confirm that there is no premature loss of c4da neurons, we additionally knocked down *cerS* in c4da neurons and investigated if all c4da neurons were present in wandering LIII larvae. Indeed, at this stage all dorsal ddaC (S2F Fig), ventro-lateral v'ada, and ventral vdaB (S2G Fig) c4da neurons could be imaged. To then ask whether c4da neurons also survive until the adult stage we imaged only the abdominal ventro-lateral v'ada c4da neurons. Those, in contrast to other c4da neurons, such as the ddaC neurons, can be visualized also in adult animals because they lie beneath the more transparent ventral side of the abdomen. Additionally, and in contrast to vdaB neurons, they do not undergo age-dependent apoptosis during the pupal stage [41]. Here, all v'ada c4da neurons were present at 5–7 days post-eclosion (N = 5 animals) (S2H Fig). Overall, these data indicate that *cerS* mutant c4da neurons exhibit aberrant dendrite morphology but do not show signs of caspase activation and the lack of CerS function does not influence c4da neuronal viability. To finally explore whether the altered c4da dendrite morphology was associated with functional response defects, we performed a mechano-nociceptive assay to measure the c4da neuronal responses to noxious stimuli [42,43]. To ensure the reduction of CerS function in all c4da neurons, we knocked down *cerS* via RNAi instead of using MARCM, which yields only a very small number of mutant da neurons per animal. Animals harboring the *UAS-cerS^dsRNA1* construct alone (*UAS-cerS^dsRNA1*/+) showed already a slight impairment in their mechano-nociceptive response compared to animals of the second control genotype (*21-7-Gal4 > UAS-Dicer2*/+) and merely a slight non-significant reduction of their nociceptive responses. However, the *UAS-cerS^dsRNA2* construct resulted in significantly fewer nociceptive responses compared to both control genotypes (Fig 1G), suggesting that *cerS* is also required to maintain c4da neuron function.

## Impairment of the Cer-PE *de novo* synthesis pathway leads to dendrite simplification in c4da neurons

To investigate how the lipid composition is altered by the lack of CerS function, we analyzed the lipidome of *cerS*-mutated animals using shotgun tandem mass spectrometry (MS). *cerS^KO* and *cerS^H215D* homozygous mutant animals die very early in development as small larvae, which morphologically correspond in size to first instar. To study the lipidome of a full body *cerS* mutant, we have therefore used the weaker *cerS^P61* allele. *cerS* mRNA is reduced by approximately 60% in this mutant, which allows a low number of slim larvae to reach the LIII stage [24]. Although the mRNA reduction is mild enough to allow escapers to develop, it is still sufficient to observe simplified dendrites in c4da neurons of wandering LIII larvae (Fig 2A). As previously reported [24], we found a reduction of the total lipid level (Fig 2B). CerSs are most prominently involved in ceramide *de novo* synthesis. By modifying their head groups ceramides can be turned into complex SL. For example, by adding phosphoethanolamine (PE) through Ceramide phosphoethanolamine synthase (CpeS) ceramides

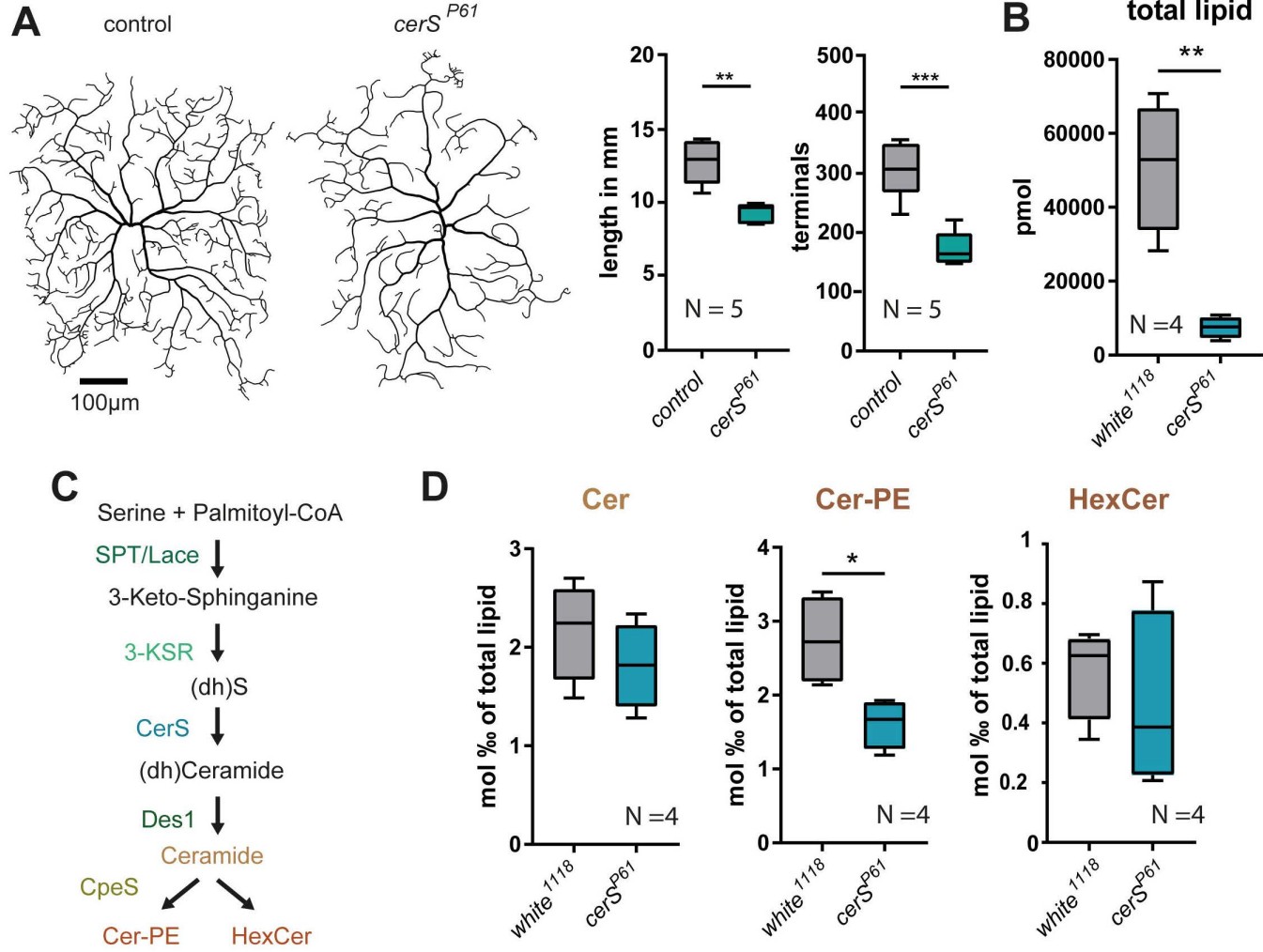

**Fig 2. Altered *cerS* expression affects total lipid and Cer-PE levels. A-D,** lipidomics were performed using the strong hypomorph *cerS^P61^*. A, C4da neurons were labeled using ppk::cd4tdTomato in *cerS^P61^* mutant and control animals. Left, representative tracings. Right, quantified total dendrite length and total number of dendritic terminal endpoints. **B-D,** Lipids were extracted from control larvae (*white^1118^*) or *cerS^P61^* mutant escapers at the LIII stage and lipid levels were measured by tandem shotgun MS. Graphs show total lipid levels (**B**) and relative ceramide, Cer-PE, and HexCer (**D**) levels. **C,** Schematic representation of the SL *de novo* synthesis pathway. Statistical test used in A, B and D: unpaired *t*-test.

are turned into Cer-PE, the major phosphosphingolipid in *Drosophila*. They can also be converted into hexosylceramides by the addition of glucose or galactose via glucosylceramide synthase (GlcT) (Fig 2C). We thus measured the relative ceramide, Cer-PE, and HexCer levels. Taking the overall low lipid levels of *cerS^P61^* mutant larvae into account we found the relative Cer-PE level to be significantly decreased (Fig 2D). We next tested how the defective Cer-PE production affects the c4da neuronal morphology and generated MARCM clones bearing mutations that block the Cer-PE producing pathway at different stages. Comparing the dendrite morphology of c4da neurons carrying loss of function mutations for *lace* (*lace^SK6^*), *des1* (*des1^KO^*), or *cpeS* (*cpeS^KO^*) with the effect of the loss-of-function allele *cerS^H215D^* we found that neurons carrying mutations at each of the steps of the Cer-PE producing pathway display reduced dendritic complexity. However, loss of *cerS* activity in c4da neurons resulted in the quantitatively strongest reduction of dendrite complexity (Fig 3A–3C). We verified this result by knocking down the Spt1 complex members *spt1* or *lace* as well as *3-ksr, cerS, des1,* or *cpeS*

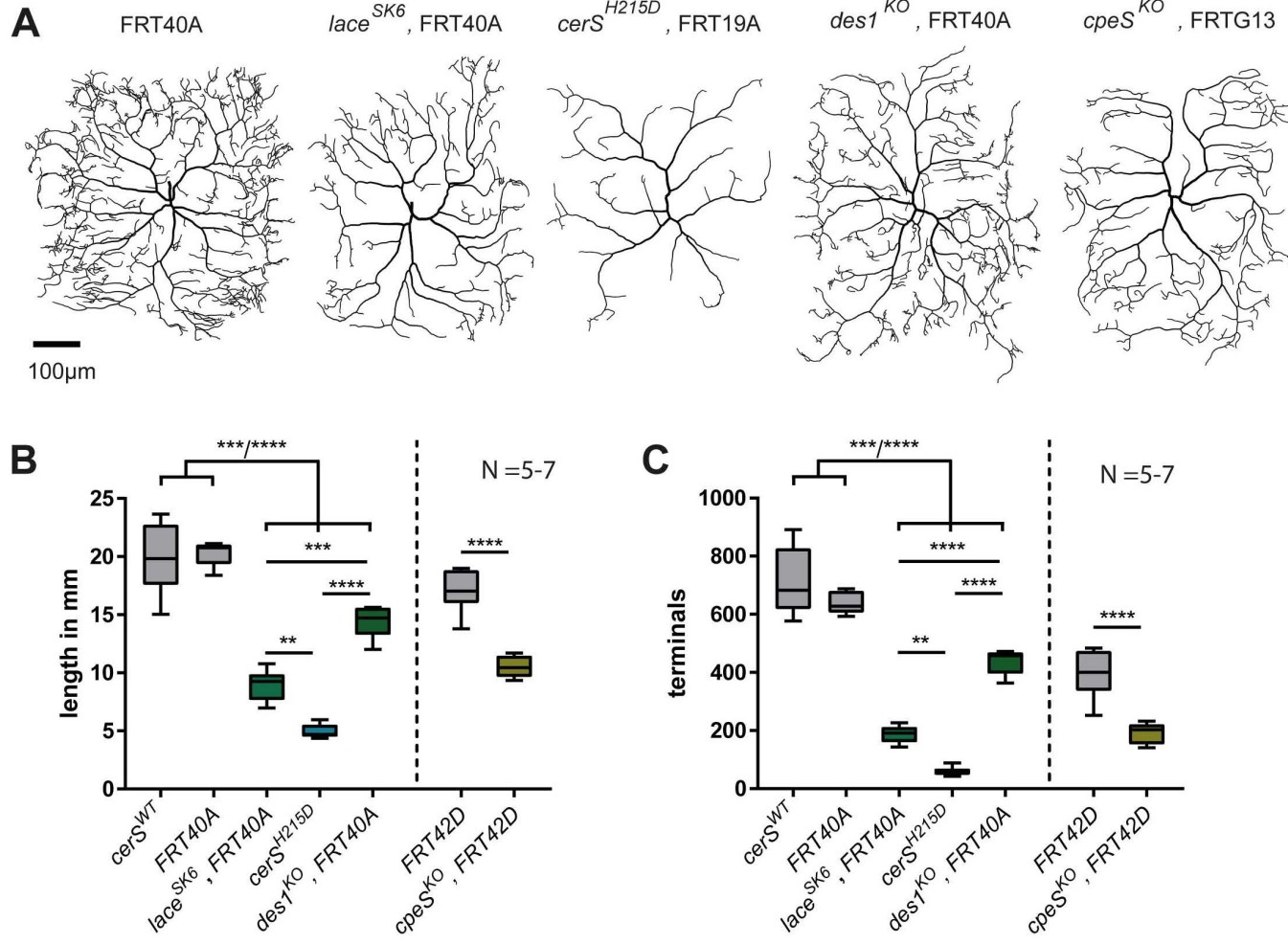

**Fig 3. *cerS* deficiency strongly affects dendrite morphology. A-C,** Single homozygous *lace^SK6^*, *cerS^H215D^*, *des1^KO^*, and *cpeS^KO^* mutant c4da neurons were generated using the MARCM technique. Representative neuronal tracings are shown in **A.** Quantified total dendrite length and total number of dendritic terminal endpoints are shown in **B** and **C**, respectively. Statistical test in B and C: 1-Way-ANOVA followed by Tukey's multiple comparison test (left); unpaired *t*-test (right).

in c4da neurons using RNAi. The results confirmed the importance of Cer-PE production for c4da neuronal morphology. By contrast, knocking down *glcT*, which catalyzes Hexosylceramide formation, did not affect c4da neuronal morphology. Similarly to the outcome of the MARCM approach, the dendrite morphology of c4da neurons was most severely affected by knocking down *cerS* (S3 Fig).

To test whether, beyond its cell-autonomous role in cd4a neurons, CerS expression in non-neuronal cells also contributes to dendrite complexity, we knocked down CerS in the epidermis and glia. Both of these cell types are in physical contact with the c4da neurons [44,45]. However, knocking down *cerS* in glia (using *repo-Gal4*) or epidermal cells (using *A58-Gal4*) left c4da neuronal morphology unaffected. These data contradict the possibility of additional requirement of CerS function in the glia or epidermis on c4da neuronal morphology (S4 Fig). Taken together, our data indicate that Cer-PE production in c4da neurons is essential for establishing complex dendritic morphology. However, the fact that impairment of enzymes upstream or downstream of *cerS* leads to milder phenotypes than impairment of *cerS* function itself suggests that the strong phenotype in *cerS* mutant neurons may be due to additional metabolic problems.

## Elevated (dh)S1P levels negatively affect dendrite morphology

The Spt-complex generates (dh)S, the substrate for the CerS reaction. Indeed, elevation of LCBs like Sphingosine, (dh)S, S1P, and (dh)S1P was previously already correlated with a decreased dendrite complexity [14–16]. LCBs are bioactive signaling molecules and are involved in signaling pathways regulating survival, apoptosis, and proliferation [23,46,47]. Given the milder phenotype of the Spt-complex member *lace* in comparison to the *cerS*[H215D] mutation, we tested if the accumulation of LCBs contributes to the effect of a *cerS* mutation on neuronal morphology (Fig 4A). To address this, we first asked which of these LCBs accumulated in *cerS* mutant fruit flies. As above, we used hypomorph *cerS*[P61] mutant larvae for lipidomics and measured increased levels of (dh)S and (dh)S1P (Fig 4B). Since the accumulation of the phosphorylated and/or of the non-phosphorylated metabolite influences neuronal viability [14,15] we tested which of the two species has greater influence on c4da dendrite morphology. To do so, we generated *cerS*[WT] and *cerS*[H215D] expressing neurons in MARCM experiments in which we simultaneously expressed sphingosine kinases (SK) which converts (dh)S into (dh)S1P (Fig 4A). The *Drosophila* genome encodes two SKs (SK1 and SK2), that display overlapping functions, as both complement a yeast SK mutant [48]. They appear to be functionally redundant in flies, since single SK1 or SK2 mutants are viable while SK1 and SK2 double mutants are lethal. Their overexpression reduces Sphingosine and (dh)S levels, while SK2 overexpression additionally enhances (dh)S1P levels [14,49]. In c4da neurons, the expression of either of the two SKs in wildtype c4da neurons caused dendrite reduction. Additionally, it strongly enhanced the loss of dendrites in *cerS*[H215D] deficient cells, suggesting a dose-dependent effect of phosphorylated LCBs on neurites (Fig 4C, 4D). We next tested whether blocking the *de novo* formation of (dh)S or (dh)S1P could rescue dendrite loss in *cerS*[H215D] mutant neurons. However, knockdown of *lace* or of *sk2* in *cerS*[H215D] mutant c4da neurons did not improve dendritic morphology. We additionally downregulated the expression of a G-protein coupled receptor (Tre1) that binds phospholipids and was recently suggested to act as a S1P receptor in *Drosophila* [50,51]. Again, no rescue could be observed (Fig 4E and 4F). By contrast, the expression of Sphingosine-1-phosphate lyase (Sply), which degrades (dh)S1P to a fatty aldehyde and phosphoethanolamine, mildly rescued the dendritic phenotype (Fig 4C–4E). Taken together, while phosphorylated LCBs have been frequently described as anti-apoptotic factors [52], our data suggest that increased levels of phosphorylated LCBs negatively affects dendrite morphology in a dose-dependent manner. However, genetic manipulations, that presumably should block LCB production, action, or degradation, only very mildly rescued the dendrite loss of *cerS*[H215D] mutant c4da neurons, if at all, suggesting that additional factors might be at play.

## C4da neurons dendrite complexity depends on C18-C24 ceramides supply

Since genetically reducing LCB levels could not, or only mildly, improve *cerS*[H215D] mutant c4da neurons, we tested if restoring ceramide/Cer-PE levels would rescue more efficiently. To test which ceramides should be used for rescue experiments, we analyzed which ceramide species are produced by the endogenous fruit fly CerS and thus may be missing in *cerS* mutant cells. Again, we used homozygous *cerS*[P61] mutant larvae. The most abundant ceramides in wild-type *Drosophila* are C20- and C22-ceramides. Additionally, much lower concentrations of C12-, C16-, C18- and C24-ceramides could be detected. Comparing the abundance of ceramides in the control and the *cerS*[P61] mutant, we found that only ceramides with acyl-chains between C18-C24 carbons, but not C12- or C14-ceramides, were reduced in the *cerS*[P61] mutant (Fig 5A). This suggests that Drosophila CerS produces ceramides with C18-C24 acyl chains and that C12- or C16-ceramides, which are found in robust amounts in mutants and controls, may originate from exogenous sources such as food or gut bacteria.

Interestingly, a shift in the ceramide species composition was also observed in *cerS1* deficient mice but the impact of an imbalance in ceramide species on the nervous system had not been investigated [15]. The differential impact of the *cerS* mutation on the abundance of ceramides of different lengths prompted us to test the effect of restoring the levels of individual ceramide species. We first tried dietary supplementation with long-chain ceramides and fed larvae carrying single *cerS*[H215D] mutant c4da cell clones with C-16-, C-18-, or C22-ceramides. Indeed, supplementation with C22-ceramides led to a mild improvement of c4da dendrite complexity (S5 Fig). In contrast, C-16-ceramides had no effect.

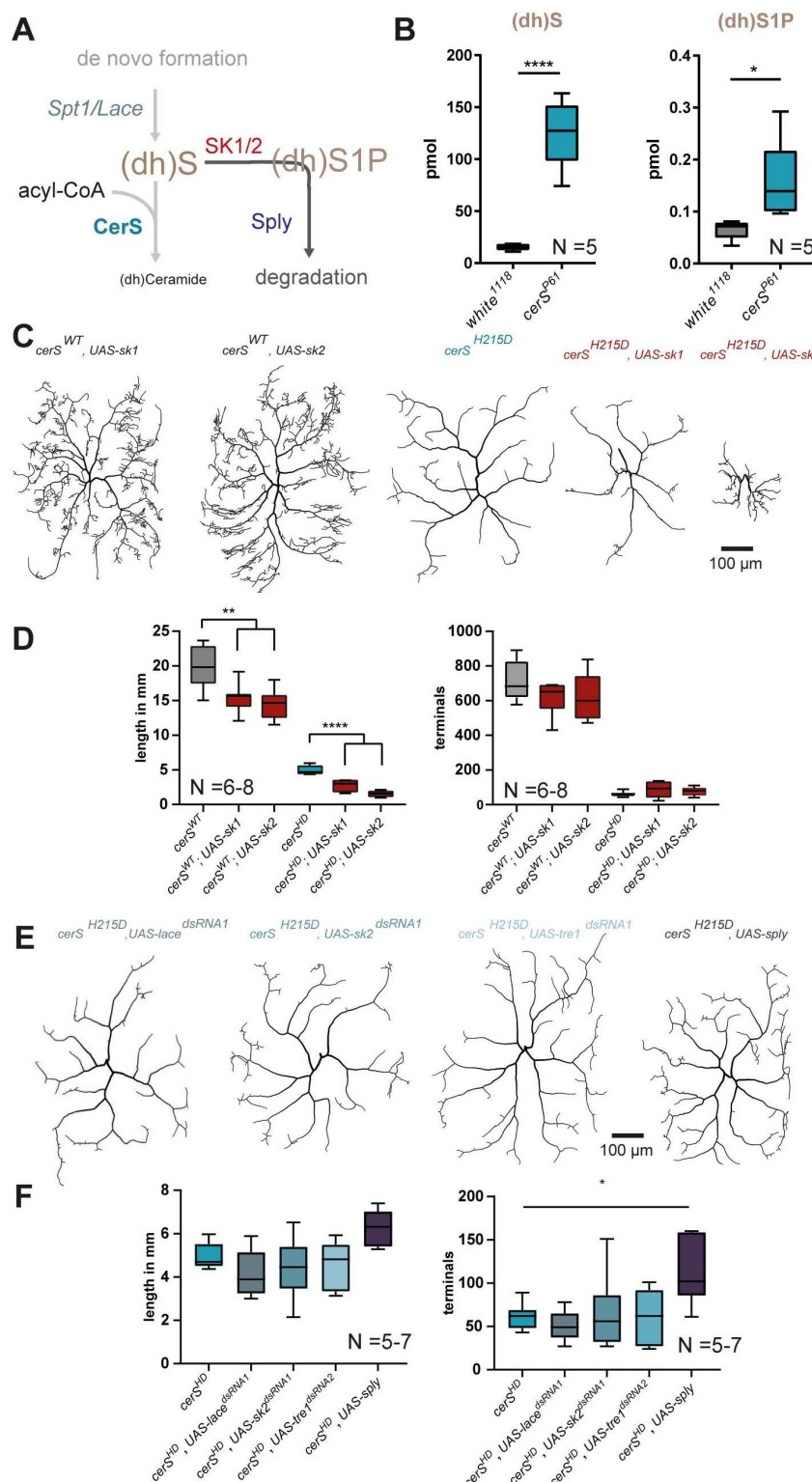

**Fig 4. Elevated (dh)S1P levels negatively alter dendrite morphology. A,** (dh)S accumulates in *cerS* mutant cells and is converted to (dh)S1P by SK1 or SK2. (dh)S1P is degraded to ethanolamine phosphate and a fatty aldehyde by Sply. **B,** absolute levels of (dh)S and (dh)S1P in control (*white1118*) and (*cerSP61*) mutant larvae. **C and D,** *cerSWT* or *cerSH215D* c4da single cell MARCM clones overexpressing SK1 or SK2. Representative neuronal tracings are shown in **C.** Quantified total dendrite length and total number of dendritic terminal endpoints are shown in **D. E and F,** homozygous *cerSH215D* c4da

MARCM clones simultaneously expressing knockdown constructs for *lace*, *sk2*, or *tre1* and homozygous *cerS^H215D* c4da MARCM clones overexpressing Sply. Representative neuronal tracings are shown in **E**. Quantified total dendrite length and total number of dendritic terminal endpoints are shown in **F**. Statistical test used in B: unpaired *t*-test and in D and F: 1-Way-ANOVA followed by Dunnett's multiple comparison.

Since our data indicates that both the accumulation of (dh)S and (dh)S1P, as well as the lack of specific ceramides for Cer-PE production (Fig 5A), are detrimental for c4da neuronal morphology we reasoned that both metabolic alterations must be reverted to obtain a full rescue of the phenotype. To this aim, we took advantage of the specificity of mammalian CerSs towards acyl-chains of a given length to produce ceramides (Fig 5B). We thus created *cerS^H215D* mutant c4da cell clones, in which we rescued the lack of the *Drosophila* CerS function by expressing different murine CerS enzymes. It is important to note that in this experiment the substrate for the CerS reaction (dh)S and its metabolite (dh)S1P are lowered and at the same time only ceramides with specific acyl chains are provided. Consistent with the lipidomics data showing that C20 and C22 ceramides are most reduced ceramide species in *cerS* mutants, only the dendritic trees of mutant neurons expressing mCerS2 or mCerS4, which produce these ceramides [19], were comparable to wild-type c4da neurons. This demonstrates that these mouse enzymes can fully compensate for the loss of the *Drosophila* enzyme. By contrast, expression of mCerS5, which preferentially utilizes C16 fatty acyl-CoA, only minimally improves the dendrite complexity of *cerS* mutant c4da neurons. Expression of mCerS1, which produces mainly C18-ceramides, improved the c4da neurons phenotype in comparison to mCerS5 overexpression. Nonetheless, total dendrite length and number of dendritic terminals of c4da neuronal dendrites were still reduced compared to control cells (Fig 5C and 5D).

To confirm that mCerS5 is functional and that it efficiently generates C16-ceramides in fruit flies, we overexpressed *UAS-mCerS5* ubiquitously using *tubulin-Gal4* and measured the ceramide species composition. These animals had significantly more C16-ceramides indicating that the murine enzyme is active and maintains its substrate specificity in the fruit fly. Also, while we did not detect C16-CerPE in wild-type flies, this lipid species was robustly detected in flies overexpressing mCerS5 (Fig 6A). To test if a shift in ceramide species composition was leading to a reduction in dendrite complexity, we overexpressed UAS-mCerS5 as well as the other UAS-mCerS constructs in c4da neurons. Only the overexpression of mCerS5 but not mCerS1, mCerS4, or mCerS2 led to reduced dendrite branching in c4da neurons, indicating that increased C16-ceramide levels are detrimental to c4da neurons (Fig 6B–6D).

In summary, a complete rescue of the mutant dendritic phenotype can only be obtained by decreasing toxic LCBs and simultaneously providing C20-C22 ceramides, the most abundant ceramide species. Thus, our data indicate that these ceramide species are key metabolites and essential for the complex dendrite elaboration of *Drosophila* c4da neurons.

## Discussion

Diseases that affect lipid metabolism, particularly those directly linked to ceramide anabolic or catabolic pathways, frequently result in neurological diseases and neurodegeneration [3]. Here, we demonstrate that CerS function is essential for the establishment and maintenance of dendrite complexity of c4da neurons. Combining genetic analysis and lipidomics, we showed that dysfunctional CerS causes multiple alterations in lipid metabolism. Using the quantitative character of c4da dendrite morphology, we defined the complex picture of lipid metabolism alterations associated with PME8-related *cerS* mutations. Our data show that in *cerS* mutant neurons, both the lack of the end-product of the ceramide *de novo* synthesis pathway (Cer-PE) and the accumulation of the CerS substrate derivative (dh)S1P are deleterious for c4da neurons. We therefore provide evidence that parallel, individual changes in lipid homeostasis have an impact on dendrite complexity in c4da neurons (Fig 7).

### Dendrite morphology is strongly affected by elevated (dh)S1P levels

A comparison of *spt* complex mutants (*spt1* or *lace*) with *cerS^HD* showed that dendrite morphology was more severely impaired in *cerS* mutants than in *spt* mutant neurons. This result was obtained using two different techniques (MARCM

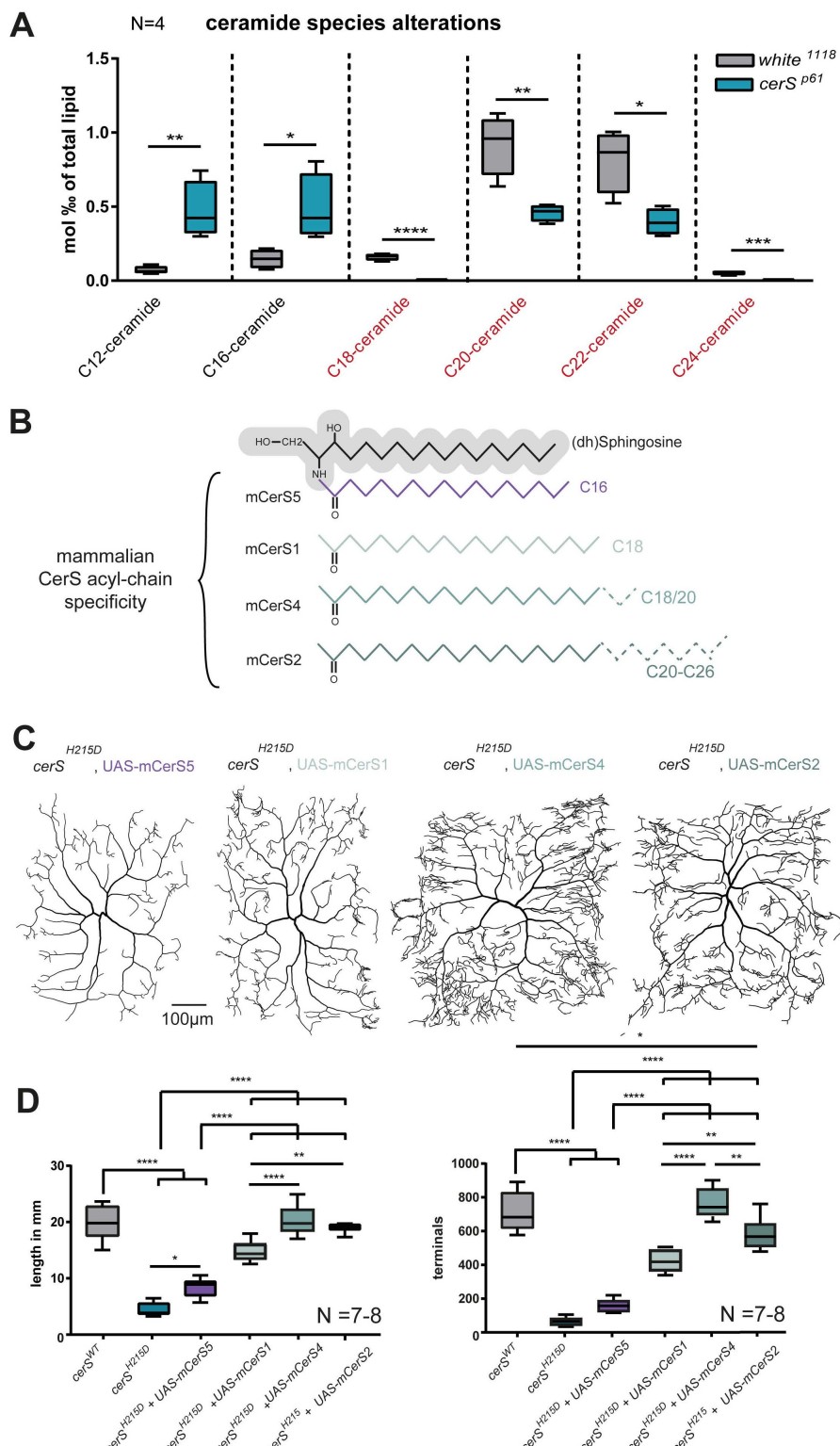

**A** N=4  ceramide species alterations

white [1118]
cerS [p61]

C12-ceramide   C16-ceramide   C18-ceramide   C20-ceramide   C22-ceramide   C24-ceramide

**B** mammalian CerS acyl-chain specificity

(dh)Sphingosine

mCerS5 — C16
mCerS1 — C18
mCerS4 — C18/20
mCerS2 — C20-C26

**C**
cerS [H215D], UAS-mCerS5   cerS [H215D], UAS-mCerS1   cerS [H215D], UAS-mCerS4   cerS [H215D], UAS-mCerS2

100μm

**D**

**Fig 5. Neuronal morphology relies on a supply with C18-C24 ceramides. A,** Lipids were extracted from *white*[1118] (control) and *cerS*[P61] (mutant) LIII larvae and the amounts of the specific ceramide species were measured by tandem shotgun MS. **B,** Schematic representation of acyl-chain substrate specificity of mCerSs. The (dh)S backbone is represented in gray and specific acyl-chains used by mCerS in color. **C-D,** mCerSs generating mostly

C16-Cer (mCerS5), C18-C20-Cer (mCerS4), or C20-C26-Cer (mCerS2) were expressed to rescue the lack of endogenous CerS function in homozygous *cerS^{H215D}* mutant c4da MARCM clones. Representative neuronal tracings are shown in **C.** Graphs show the quantified total dendrite length and total number of dendritic terminal endpoints **(D)**. Statistical test used in A: unpaired *t*-test and in D: 1-Way-ANOVA followed by Dunnett's multiple comparison.

and RNAi). The SPT-complex is upstream of CerS in the ceramide *de novo* synthesis pathway and produces LCBs which are converted into (dh)ceramide by CerS. The difference between SPT-complex and CerS mutants in our study may be explained by the fact that *spt*-complex mutants are mostly suffering from a reduction of complex SL. On the other hand, *cerS^{HD}* mutants suffer from a reduction in complex SL and increased LCB levels.

Modified LCB levels were frequently linked to neuronal dysfunction and altered neuronal morphology [17,53,54]. Interestingly, not only high but also low levels of LCBs were correlated to morphological defects. For example, a lack of LCB production in SPT-complex mutant *Drosophila* neurons was correlated with protein sorting defects and aggregation of the cell recognition protein Down syndrome cell adhesion molecule (Dscam) [53]. Dscam regulates multiple processes in *Drosophila* neurons such as proper axonal morphology, correct axonal innervation of target regions, and dendritic self-avoidance [32,53,55]. The link between SPT-complex activity and morphological defects was investigated using the *Drosophila* adult Mushroom body (MB), a neuropil in the central nervous system [53,56]. During development, the absence of SPT-complex-generated sphingolipids in MB neurons led to an aggregation of Dscam. Axon and dendrite specific Dscam isoforms were not properly distributed into their target compartments which resulted in a failure to segregate axonal branches into distinct MB lobes [53]. However, it remains unclear whether the lack of LCBs or a lack of complex SL is causing Dscam-mediated axon sorting defects.

Nevertheless, multiple studies including ours clearly correlate elevated LCB levels to neurodegeneration [11,17,34]. Additionally, a genotype-phenotype association study of Hereditary sensory and autonomic neuropathy type 1 (HSAN1) in patients also pointed towards a highly toxic effect of high LCB levels on the nervous system. HSAN1 is caused by alterations in the SPT-complex that acts upstream of CerS in the ceramide *de novo* synthesis pathway. So far, 17 point mutations have been mapped. The best-studied point mutations change the substrate specificity of the SPT complex. This complex typically condensates serine with acyl-CoA but these mutations allow it to efficiently utilize L-alanine and glycine which then forms atypical LCBs. Such 1-deoxysphinganine and 1-(deoxymethyl)sphinganine lack the hydroxy group which serine would provide. The serine-derived hydroxy group in typical LCBs is critical for a conversion into complex SL and also for degradation. Thus, its lack prevents 1-deoxisphingoloipids not only from being converted into complex SLs and also to be degraded through the S1P-Lyase pathway [10,11,34]. Of greater interest to our study, three of the 17 above-mentioned mutations increase the enzymatic activity. This results in an increase in common LCBs, such as (dh) Sphingosine. Patients carrying these mutations show exceptionally severe HSAN1 symptoms. Therefore, high plasma LCB levels were suggested as a biomarker that could help to predict the severity of the HSAN1 [10].

Naturally occurring LCBs can be phosphorylated to form S1P and (dh)S1P and it was unclear which LCB species is most detrimental to neuronal viability. We first demonstrated that both (dh)S and (dh)S1P levels were increased in *cerS* mutants. To clarify which of those is more toxic, we expressed SK1 or SK2 in either wildtype or *cerS^{H215D}* mutant neurons and found that in both cases the additional expression of SKs reduced dendrite branching. *cerS^{H215D}* mutant neurons that simultaneously expressed SKs should have had the highest levels of phosphorylated LCBs. Since their dendrite morphology was affected the most, it could be suggested that (dh)S1P is the more toxic species (Fig 4). The effect of (dh)S1P had not yet been explored extensively in neurons, but its analog S1P has been shown to act on neural development, differentiation, migration, survival, and synaptic transmission [57,58]. It is therefore not surprising that its concentration is tightly regulated by SKs and phosphatases like Sply. Phosphorylated LCBs are pleiotropic messenger molecules that can act through S1P-receptors [59]. Blocking this signaling pathway might represent a valuable approach to at least reduce the loss of dendrites.

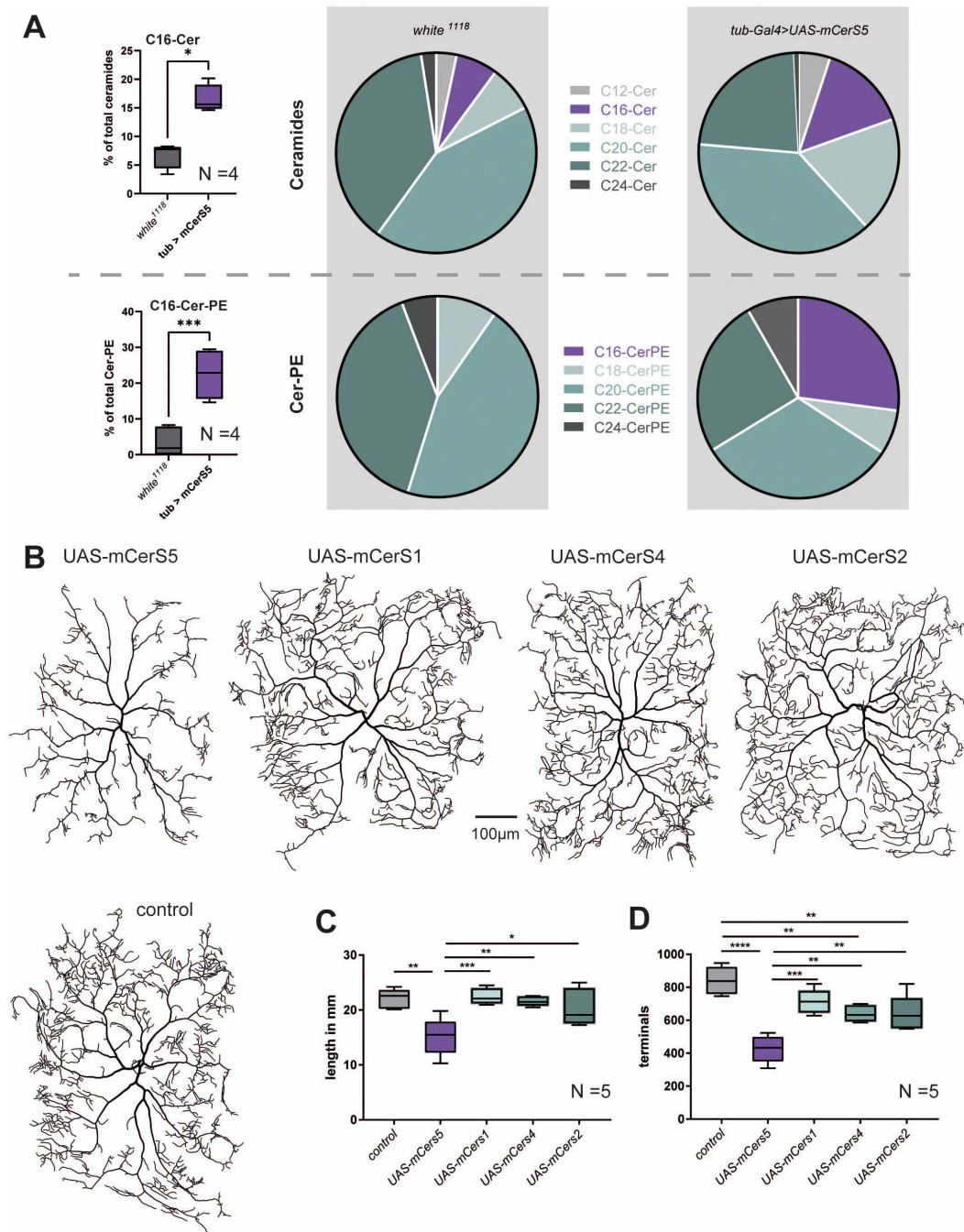

**Fig 6. A *UAS-cerS5* was ubiquitously overexpressed using *tub-Gal4*.** Lipids were extracted from *white[1118]* (control) and mCerS5 overexpressing LIII larvae and ceramide levels were measured by tandem shotgun MS. Graphs show quantified levels of C16-Ceramide and C16-Cer-PE **(left)**. Pie diagrams show ceramide and Cer-PE species distribution (right). **B-D**, *ppk-Gal4* was used to overexpress *UAS-mCerS5, UAS-mCerS1, UAS-mCerS4*, and *UAS-mCerS2* in c4da neurons. Representative neuronal tracings are shown in **B**. Graphs show the quantified total dendrite length and total number of dendritic terminal endpoints **(C, D)**. Statistical test used in A: Mann-Whitney test and C/D 1-Way-ANOVA followed by Dunnett's multiple comparison.

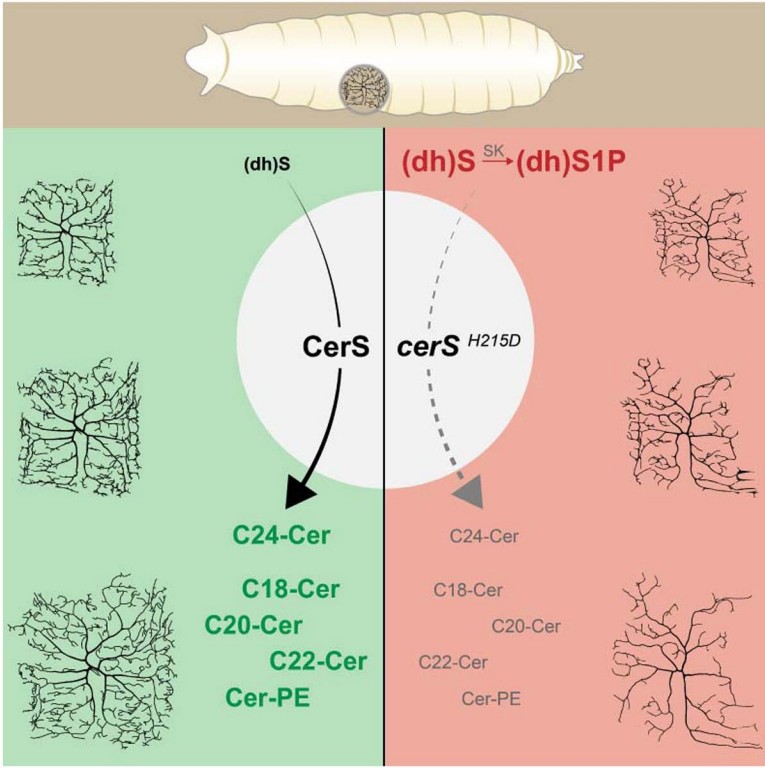

**Fig 7. Model.** CerS activity reduces LCBs such as (dh)S and (dh)S1P while producing long-chain ceramides (left). In *cerS* deficient neurons (dh)S and (dh)S1P levels are elevated and simultaneously C18-C24-ceramides diminished. To fully rescue neuronal vitality, both metabolic alterations must be reverted.

Toxic lipids including the above-mentioned LCBs can induce a variety of signaling pathways, many of which converge in caspase-dependent apoptosis. Our study excluded caspase's role in early dendrite loss, but numerous non-caspase-mediated signaling pathways including ferroptosis or autophagic cell death could be at play. In future, it will be important to identify the signaling pathways triggered in this specific context.

### Lack of complex sphingolipids affects c4da dendrite morphology

In addition to the accumulation of LCBs, our study showed that the *cerS* mutation also leads to a strong reduction in Cer-PE levels. Cer-PE is the most abundant complex SL in invertebrates and the functional counterpart of mammalian sphingomyelin (SM). It is produced from ceramide by CpeS [20,60]. In *Drosophila*, the lack of glial Cer-PE production in *cpeS* mutants was recently shown to cause photosensitive epilepsy and a perturbed circadian rhythm. *cpeS* deficiency during development resulted in animals that had morphologically defective glial cells and a rescue of *cpeS* expression in glia was able to rescue the aforementioned defects [61,60].

SLs are components of membranes and are therefore important for dendrite growth during development. Together with glycosphingolipids and cholesterol, complex SL (such as Cer-PE or SM) also form lipid rafts with higher viscosity at the plasma membrane [61–63]. Lipid rafts are required for the clustering of proteins or signaling molecules, for membrane trafficking, and interact with the submembrane actin cytoskeleton [63]. A Cer-PE deficient nervous system fails to establish lipid rafts [61] and it is highly likely that all above mentioned processes might well be altered in *cerS* deficient neurons. This could also contribute to the aberrant c4da dendrite morphology.

## Neuronal morphology relies on C18-C24 ceramides supply

To gather information towards an efficient therapy of PME8, we have carried out various rescue experiments. However, when we tried to solely rescue the lack of complex SL by feeding ceramides or to prevent only the accumulation or action of LCBs, we observed only very mild rescue effects, if any. The most efficient way to lower LCBs and enhance complex SL was therefore the re-expression of CerS enzymes in cerS^H215D mutant neurons. However, we found that only the expression of mCerS4 or mCerS2, which produce ceramides with acyl-chains that resemble the ceramides produced by fly CerS, fully restore dendrite branching.

When comparing the composition of ceramide species between the mutant and the control, we found that *cerS* mutants have an increased relative level of C12- and C16-ceramides, while C18-C24-ceramides were reduced. The relative abundance of each ceramide species was altered. We propose that the fly CerS is responsible for the synthesis of longer acyl-chain ceramides (C18-C24), while C12- and C16-ceramides are derived from an exogenous source such as the diet or gut bacteria [64,65]. Interestingly, also shown in *cerS1* and *cerS2* mutant mice, increased C16-ceramides levels were observed [15,16,21,66]. Thus, an imbalance in the relative abundance of ceramide species might be a conserved trait. However, it remains unclear whether the shift in ceramide species composition also contributed to the observed loss of Purkinje cell neurites in these mutant mice [16,67]. Indeed, attention has only recently been paid to the effects of individual ceramide species, such as C16-ceramides, on cell vitality and survival [21,68]. C16-ceramides can induce ER stress [22,68–70]. Additionally, in mitochondria, C16-ceramides may form channels permeabilizing the mitochondrial outer membrane leading to mitochondrial dysfunction [68,71,72].

We not only showed the differential impact of a *cerS* mutation on the abundance of different ceramide species, our analysis of cerS^H215D mutant c4da neurons allowed us also to dissect the role of individual ceramide species. We found that only CerS enzymes that synthetize long-chain (C20-C24-) ceramides can efficiently restore dendrite complexity. This finding supports the view that individual ceramide species display distinctive functions. It furthermore allows to hypothesize a path for improving the outcome of *cerS* mutations, including in PME8 patients. While our genetic analysis pointed to a cell autonomous requirement for ceramide synthesis in c4da neurons, we also found that mere dietary supplementation of the appropriate long-chain ceramides could improve the c4da neurons defects. We thus envision that as an alternative to the re-expression of CerSs, a combination of reducing the effect of toxic LCBs, such as (dh)S1P, together with a supplementation of long-chain ceramides will be an important path to be tested in future experiments in vertebrate models of PME8.

In summary, we showed that the dendrites of *cerS* deficient neurons display reduced complexity already early in development and additionally lose dendritic branches at later developmental stages. We dissected each of the multiple alterations in lipid metabolism that arise in *cerS* deficient cells and analyzed their individual quantitative impact on the morphology of c4da neurons. Overall, our analysis indicates that the accumulation of LCBs, in particular (dh)S1P, and the reduction of long acyl-chain ceramide species (C18-C24-ceramides) are the key factors determining the neuronal morphology, highlighting potential strategies to restore neuronal integrity in this model of PME8 (Fig 5). This work demonstrates the importance of a detailed lipidomic analysis combined with genetic dissection of metabolic pathways to formulate informed hypotheses on the origin and potential treatment of lipid-metabolism related diseases.

## Materials and methods

### Fly husbandry and maintenance

The exact genotypes used in the experiments are listed under S1 Table.

To generate MARCM clones, mutant *cerS* alleles cerS^H215D (this work), cerS^KO and the control allele *CerS^WT* [35] were recombined with the FRT19A site. lace^SK6, FRT40A was obtained from the NIG stock center (M2L-0156). This allele lacks 11 bp, creating a frame shift and a truncated protein. des1^KO, FRT40A [73] was generously provided by Chih-Chiang Chan

(National Taiwan University). *cpes^KO*, FRT42 [61] was generously provided by Usha Acharya, University of Massachusetts. MARCM clones were generated by using the SOP-FLP lines Nr. 109946 (FRT19A), 109947 (FRT40A), and 109949 (FRT42D) (Kyoto stock center, Japan) [74].

*white^1118* (Nr. 3605), w^67c23,P{lacW}schlankG0061/FM7c (Nr. 11665), FRT19A (Nr. 1744), FRT42D (Nr. 1802), FRT40A (Nr. 8212) and UAS-CD4tdTomato (Nr. 35844) were obtained from the Bloomington Stock Center (Bloomington, USA).

For RNAi experiments we generated a fly line harboring *ppk-Gal4* (Chr. III) (kind gift from Yuh-Nung Jan, Howard Hughes Medical Institute, San Francisco, USA) recombined with *UAS-mCD8GFP* (Bloomington Stock Center,Bloomington, USA, Nr. 5130) and *UAS-Dicer2* (Chr. II, kind gift from Justin Blau, New York University, USA) and crossed it to the following UAS-RNAi lines: *UAS-cerS^dsRNA1* (Nr. 109418); *UAS-cerS^dsRNA2* (Nr. 41115); *UAS-des1* (Nr. 106665); *UAS-lace^dsRNA1* (Nr. 21805); *UAS-lace^dsRNA2* (Nr. 110181); *UAS-spt1^dsRNA1* (Nr. 10020); *UAS-spt1^dsRNA2* (Nr. 108833); *UAS-3KSR^dsRNA1* (Nr.6734); *UAS-3KSR ^dsRNA2* (Nr. 109284); *UAS-OR42b^dsRNA* (control, Nr. 101143); *UAS-cpes ^dsRNA1* (Nr. 3594); *UAS-cpes ^dsRNA2* (Nr. 102245); *UAS-glcT^dsRNA1* (Nr. 44912), *UAS-sk2^dsRNA1* (Nr.101018) all obtained from the VDRC stock center in Vienna, Austria. *UAS-glcT ^dsRNA2* (Nr. 67304) and *UAS-tre1 ^dsRNA1* (Nr. 34956) were obtained from the Bloomington Stock Center (Bloomington, USA). Crosses were kept at 27 °C.

*UAS-cerS* (Nr. F000075) was obtained from the FlyORF collection Zürich Switzerland, *UAS-SK1* and *UAS-SK2* [14] were kindly provided by Usha Acharya, University of Massachusetts Medical School, US. *UAS-spin* (Nr. 39669), *UAS-apoliner* (Nr.32121) and *UAS-p35* (Nr. 5072 and Nr. 5073) were obtained from the Bloomington Stock Center (Bloomington, USA), and *UAS-sply* [75] was generously provided by Corinne Antignac, Paris, France.

*ppk-mCD8GFP* (Chr. 3, [45]) was a kind gift from Yuh-Nung Jan (Howard Hughes Medical Institute, San Francisco, USA). *A58-Gal4* [76] was a kind gift from S. Rumpf (University of Münster, Germany). *Tub-Gal4* was obtained from the Bloomington Stock Center (Bloomington, USA). (Nr. 5138). *21–7-Gal4* was a gift from YN Jan, SF, USA.

Flies were maintained on standard medium at 25°C, unless otherwise stated. Ceramides for the feeding assay were purchased from Avanti Research, US, and used at a concentration of 0,16 mg/ml (C16-ceramide) and 0,17 mg/ml (C18-ceramdieand C22-ceramide).

## Generation of the *cerS^H215D* mutant

*cerS* exons 2–6 were cloned into pGE-attB-*cerS^WT* by Sociale et al 2018 [36,77]. To generate pGE-attB-*cerS^H215D* quick change PCR was used to replace histidine at position 216 by aspartate using pGE-attB- *cerS^WT* and the following primers: cagatgttcatc**gat**cacatggtcacc and ggtgaccatgtg**atc**gatgaacatctg (bases encoding for histidine/aspartate are indicated in bold). *cerS^H215D* transgenic flies *were* made as previously described. In brief, we used the previously published founder line (*cerS^KO*) in which *cerS* exons 2–6 were removed by homologous recombination while simultaneously introducing an attP landing site [36,77]. Next, the landing site was used to re-integrate the *cerS* encoding exons via ΦC31-mediated integration. The integration of pGE-attB-white^+-*cerS^H215D* into the white eyed *cerS^KO* was done by BestGene Inc. (Chino Hills, CA, USA) and transgene integration was marked by White^+ expression. The correct re-insertion of the *cerS* encoding sequence was tested by PCR. Since *cerS^H215D* expression is derived from the endogenous locus it should follow the expression level of wildtype *cerS* which we confirmed via qRT-PCR.

## Generation of the UAS-mCerS constructs

cDNA clones were obtained: for CerS1 from OriGene Technologies, Rockville, USA (Cers1 (NM_138647); for CerS5/Lass5 from imaGenes (IRAVp968B03129D full length cDNA clone in pCMV-SPORT6 vector); for CerS2/Lass2 from imaGenes (IRAVp968E1116D mouse full length cDNA clone in pCMV-SPORT6 vector). The cDNAs were PCR amplified using the following primers: CerS5 ATATGCGGCCGCATGGCGACTGCAGCAGCGGAAACC and CCTACCTCTAGACTAGTCACAGGAGTGTAGATGTGGGGAGGC; CerS1 GAGGAGATCTGCCGCCGCGATCGCCATGGCT and GGGCTCTAGATTTAAACCTTATCGTCGTCATCCTTGTA; CerS4 (a gift of Klaus Willecke) and was excised from the donor plasmid using XhoI

and XbaI. mCD8-GFP sequence (1.4 kb) was removed from pJFRC7-20XUAS-IVS-mCD8::GFP (Addgene) by using BglII and XbaI. PCR amplified cDNAs were inserted into the modified pJFRC7 vector using XhoI and XbaI (CerS4) or NotI and XbaI (CerS5), or BglII and XbaI (CerS1) respectively. All constructs were confirmed by sequencing (Eurofins Genomics; Ebersberg, GER) and by BestGene (Chino Hills, USA) into flies carrying the attp40 (*UAS-cerS1* and *UAS-cerS4*) or attp2 (*UAS-cerS2*) landing sites.

## Live imaging and image processing

The dendritic morphology of da neurons was observed by immobilizing living larvae between a glass slide and a cover-slip in a mixture of halocarbon oil and ether (4:1). One da neuron per animal was imaged by confocal microscopy (Zeiss LSM700 or LSM710). Images were hand traced and analyzed using the TREES toolbox plug-in for MATLAB (R2014b) [78].

## Lipidomics

Lipid extraction: Lipids were extracted from individual larvae. Each larva was homogenized in 150µL water (LC-MS grade). To every homogenate, 1000µL Extraction mix (5/1 MeOH/CHCl$_3$ containing internal standards. PE[31:1] 420pmol; PC[31:1] 792pmol; PS[31:1] 197pmol; PI[34:0] 169pmol; PA[31:1] 112pmol; PG[28:0] 103pmol; CL[56:0] 57pmol; LPA[17:0 79pmol; LPC[17:1] 70pmol; LPE[17:0] 76pmol; Cer[17:0] 64pmol; SM[17:0] 198pmol; GlcCer[12:0] 110pmol; GM3[18:0-D3] 29pmol; TG[50:1-D4] 2351pmol; CE[17:1] 223pmol; DG[31:1] 128pmol; MG[17:1] 207pmol; Chol[D6] 1448pmol; Car[15:0] 91pmol; Sph[d18:0-D7] 64pmol) were added and tubes were sonicated for 30 minutes in a bath sonicator. After 2 minutes centrifugation at 20000g, supernatant was transferred to a fresh Eppendorf tube. 200 µL chloroform as well as 600µL 1% acetic acid were added to each tube, tubes were shaken manually for 10 seconds followed by centrifugation for 5 minutes at 20000g to separate phases. The lower phase was transferred to a fresh tube and tubes were dried in a speed-vac for 15 minutes at 45°C. Dried samples were redissolved by adding 1mL of spray buffer (8/5/1 isopropanol/methanol/H$_2$O (all LC-MS grade) + 10mM ammonium acetate + 0.1% acetic acid (LC-MS grade)) to each sample and sonicating for 5 minutes in a bath sonicator. Mass spectra were recorded on a Thermo Q-Exactive Plus spectrometer equipped with a standard heated electrospray ionization (HESI) II ion source for shotgun lipidomics. Samples were sprayed at a flow rate of 10 µl min$^{-1}$ in spray buffer. MS1 spectra (res. 280000) were recorded as segmented scans with windows of *m*/*z* 100 between *m*/*z* 240 and 1,200 followed by MS2 acquisition (res. 70000) from *m*/*z* 244.3364 to 1,199.9994 at *m*/*z* 1.0006 intervals. Raw files were converted into mzml files and analyzed using the LipidXplorer (version 1.2.8) [79]. Internal standard intensities were used to calculate absolute amounts, and all identified lipids were normalized to total lipid amount.

LCB levels measurements were carried out as contract work by the company Lipotype (Dresden, Germany). Data represent C14(dh)S plus C16(dh)S and C14(dh)S1P plus C16(dh)S1P of single animals. Because *cerS^P61* are slimmer and smaller than control animals all values were normalized to the wet weight.

## Mechanonociception assays

Mechanonociception experiments were performed on staged 96 h AEL ± 3 h old 3rd instar larvae with a calibrated von-Frey-filament (50 mN). Larvae were carefully transferred with a wet brush to a 2% agar plate containing a 1 ml water film. Larvae were stimulated on mid-abdominal segments (A3-A5) twice within 2 s. Behavioral responses (non-nociceptive, bending, rolling/multiple rolling) to both stimuli were noted and only behavioral responses to the second stimulus were scored and plotted. Staging and experiments were done in a blinded and randomized fashion.

## Statistical analysis

All datasets were analyzed using Prism 5.0d or 10.4.2 (GraphPad) and tested for normal distribution prior to statistical analysis. A minimal dataset containing single values, SEM, and the statistical tests used to analyze the dataset can be found in S2 Table.

## Supporting information

**S1 Fig. CerS activity is necessary for proper dendrite morphology in c1da and c3da neurons. A,** representative image of a heterozygote *cerS^KO* mutant c4da neuron labeled by pkk::mCD8GFP. **B, C:** Mutant c1da (**B**) and c3da (**C**) single cell clones expressing wild-type CerS (*cerS^WT*) or no CerS (*cerS^KO*) were generated using the MARCM technique. *cerS* expression was cell-autonomously rescued in *cerS^KO* mutant neurons using a *UAS-cerS* expression construct. Images show representative tracings of single cell MARCM clones. Graphs show the quantified total dendrite length and total number of dendritic terminal endpoints, respectively. Statistical test used: 1-Way-ANOVA followed by Tukey's multiple comparison test despite the data for c1da neurons "length" were analyzed using the Kruskal-Wallis test followed by a Dunn's multiple comparison test. **D,** Dendrites of mutant (*cerS^KO*) and control (*CerS^WT*) c4da neuronal MARCM clones were imaged at the second (LII), the feeding third (LIIIf), and the wandering third instar larval stage (LIIIw). Graphs show the quantified total dendritic length and the total number of terminal endpoints. Statistical test used: 2-Way-ANOVA followed by Sidak's multiple comparison test. **E,** *cerS^KO* c4da neuronal MARCM were imaged in early LIIIf stage larvae and 24h later. Images show representative dendritic trees and graphs the quantified total dendrite length and total number of dendritic terminal endpoints, respectively. Statistical test used in D: paired *t*-test.
(EPS)

**S2 Fig. Caspases are not activated and the c4da cell number is not affected in *cerS* mutant neurons. A,** *cerS^KO* mutant MARCM clones were labeled by *UAS-CD4tdTomato* (red) and *UAS-Apoliner* expression is shown in green. Absence of nuclear GFP localization (arrow) indicates the lack of caspase activity. **B and C,** expression of the caspase inhibitor p35 in *cerS^KO* mutant MARCM clones does not rescue of total dendritic length (B) or number of total dendritic endpoints (C). **D and E,** *cerS^H215D* mutant (**D**) and wild-type (**E**) dorsal c4da neuronal MARCM clones in late stage (P14/P15) pupa. **F-H,** *cerS* was knocked down using *ppk-Gal4* driven *UAS-cerS^dsRNA1* and neurons were labeled using ppk::CD4GFP. Dorsal (**F**) and ventral (**G**) whole body images were taken from three individuals and the presence of c4da neurons observed. Pictures show representative confocal images. **H,** the image shows a representative confocal image of ventro-lateral abdominal v'ada c4da neurons of a 5–7-day old adult fly. Statistical test used in B and C: 1-Way-ANOVA followed by Dunnett's multiple comparison test.
(EPS)

**S3 Fig. *cerS* knock-down strongly affects dendrite morphology.** Knockdown of *spt1*, *lace, 3-KSR*, *cerS*, *des1*, *ceps,* and *glcT* in c4da neurons was obtained using the c4da neuron specific ppk-Gal4 driver line and, if available, two independently generated UAS-dsRNA lines were used. C4da neurons were visualized by simultaneous co-expression of UAS-mCD8GFP. Representative dendrite tracings are shown for each gene. Graphs show quantified total dendrite length and total number of dendritic terminal endpoints. Statistical test used: Kruskal-Wallis with Dunn's multiple comparison test.
(EPS)

**S4 Fig.** A-C, *cerS* was knocked down in glia using *repo-Gal4* or epidermal cells using *A58-Gal4* and the morphology of c4da neurons was observed by ppk::CD4GFP expression. Images show representative dendritic trees (A). Graphs show quantified total dendrite length and total number of dendritic terminal endpoints (B/C). Statistical test used in B and C: 1-Way-ANOVA followed by Dunnett's multiple comparison.
(EPS)

**S5 Fig.** Dendrite defects are only mildly rescued by a ceramide enriched diet. A/B, Larvae harboring *cerS^H215D* mutant MARCM clones were raised on either C16-, C18- or C22-ceramide enriched food. Larvae harboring *CerS^WT* expressing MARCM clones were raised on regular food. Representative neuronal tracings are shown in A. Quantified total dendrite length and total number of dendritic terminal endpoints are shown in B. Statistical test used in B: Kruskal-Wallis

test followed by a Dunn's multiple comparison test (length) 1-Way-ANOVA followed by Dunnett's multiple comparison (terminals).
(EPS)

**S1 Table.** Genotypes used.
(DOCX)

**S2 Table.** Minimal data set.
(DOCX)

## Acknowledgments

We are grateful to Usha Acharya (University of Massachusetts Medical School, Worces, US) for sharing UAS-SK1 and UAS-SK2 and *cpes*^KO flies, Corinne Antignac (Institute Imagine, Paris, France) for UAS-Sply flies and Chih-Chiang Chan (National Taiwan University for *des1*^KO flies. We thank Damian Demarest, Pia Bayer, and Rachel Lazar for their help with experiments and tracings. We thank Mario Werner and Judith Reisen for helping with cloning of UAS-CerS2, UAS-CerS-4, and UAS-CerS5 transgenes. We thank Karolina Doubkova and Jonathan Smart for critical reading of the manuscript; Rita Kerpen, Regina Hube, Astrid Fleige, and Franka Eckhardt for Technical Assistance; and Nina Knubel for her assistance with the figures.

## Author contributions

**Conceptualization:** Anna B. Ziegler, Gaia Tavosanis.

**Data curation:** Anna B. Ziegler.

**Formal analysis:** Anna B. Ziegler, Konstantin Beckschäfer, Neena Dhiman.

**Funding acquisition:** Anna B. Ziegler, Gaia Tavosanis.

**Investigation:** Anna B. Ziegler, Cedrik Wesselmann, Konstantin Beckschäfer, Neena Dhiman.

**Methodology:** Anna B. Ziegler.

**Project administration:** Anna B. Ziegler.

**Resources:** Anna B. Ziegler, Anna-Lena Wulf, Reinhard Bauer.

**Supervision:** Anna B. Ziegler, Peter Soba, Christoph Thiele, Reinhard Bauer, Gaia Tavosanis.

**Validation:** Anna B. Ziegler.

**Visualization:** Anna B. Ziegler.

**Writing – original draft:** Anna B. Ziegler, Gaia Tavosanis.

**Writing – review & editing:** Anna B. Ziegler, Peter Soba, Reinhard Bauer, Gaia Tavosanis.

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
