## [Decision Letter · Decision Letter 0]

24 Jan 2025

PGENETICS-D-24-01365

Individual lipid alterations at the origin of neuronal Ceramide Synthase defects

PLOS Genetics

Dear Dr. Ziegler,

Thank you for submitting your manuscript to PLOS Genetics. After careful consideration, we feel that it has merit but does not fully meet PLOS Genetics's publication criteria as it currently stands. Therefore, we invite you to submit a revised version of the manuscript that addresses the points raised during the review process.

Please submit your revised manuscript within 60 days . If you will need more time than this to complete your revisions, please reply to this message or contact the journal office at plosgenetics@plos.org. Please include the following items when submitting your revised manuscript:

We look forward to receiving your revised manuscript.

Kind regards,

Cathy Savage-Dunn

Academic Editor

PLOS Genetics

Hua Tang

Section Editor

PLOS Genetics

Aimée Dudley

Editor-in-Chief

PLOS Genetics

Anne Goriely

Editor-in-Chief

PLOS Genetics

**Journal Requirements:**

1) Please provide an Author Summary. This should appear in your manuscript between the Abstract (if applicable) and the Introduction, and should be 150-200 words long. The aim should be to make your findings accessible to a wide audience that includes both scientists and non-scientists. Sample summaries can be found on our website under Submission Guidelines:

https://journals.plos.org/plosgenetics/s/submission-guidelines#loc-parts-of-a-submission

3) We note that your Data Availability Statement is currently as follows: "Our data will be openly available". Please confirm at this time whether or not your submission contains all raw data required to replicate the results of your study. Authors must share the “minimal data set” for their submission. PLOS defines the minimal data set to consist of the data required to replicate all study findings reported in the article, as well as related metadata and methods (https://journals.plos.org/plosone/s/data-availability#loc-minimal-data-set-definition).

- The points extracted from images for analysis..

4) Please ensure that the funders and grant numbers match between the Financial Disclosure field and the Funding Information tab in your submission form. Note that the funders must be provided in the same order in both places as well.

**Reviewers' comments:**

Reviewer's Responses to Questions

**Comments to the Authors:**

Reviewer #1: PLOS genetics comments:

This manuscript by Ziegler et al studies the impact of sphingolipid metabolic change associated with the epilepsy-associated PME8 variants of Ceramide Synthase (CerS).

The authors employed live imaging and genetic rescue experiments to pinpoint the sphingolipid metabolites in CerS mutant in Drosophila. Their results indicate that the reduced CerPE or C18-C24 ceramide in CerS mutant leads to impaired dendritic morphology. Notably, the accumulation of (dh)S1P also contributes to the dendritic abnormalities observed in these mutants. These findings enhance our understanding of the mechanism underlying sphingolipid dysregulation associated with PME8.

Also in the manuscript, although detailed analyses of the changes in sphingolipid metabolites in CerS mutants were provided, the relationship among these changes was not clarified. Specifically, the lack of CerPE, the lack of C18-C24 ceramide, and the accumulation of (dh)S1P was noted, but the mechanistic connection of these findings was largely missing. It would help the readers if the relationship of these observations (the lack of CerPE, the lack of C18-C24 ceramide, and the accumulation of (dh)S1P) were somehow experimentally explored by using genetic double mutant analyses.

Comments:

1. In Fig1, does CerS activity only affect dendritic morphology of da neurons? How about other types of neurons? In other words, is “CerS mutation causes dendritic defect” a general or da-neuron-specific issue?

2. This leads to the next comment: For general audiences, it will be helpful to explain in the introduction why the dendritic morphology of the c4da neuron is a suitable model to study PEM8.

3. What’s the phenotype of CerSP61 allele? In Fig2 and Fig3, it would help to provide its dendritic morphology, so as to complement the lipidomic analysis of this mutant.

4. Figure 4: it is assumed that mammalian CerS4 and CerS2 can restore dendritic morphology in the CerSH215D mutant via producing C18-C24 ceramide. However, lipidomic evidence was missing to show that medium-long chain ceramide was changed. This is just an example, but throughout Figures 2 to 4, the genetic experiments were conducted under the assumption that changes of specific sphingolipid species would happen. To validate that all genetic manipulations of enzymes were effective, it is advised that lipidomics be performed.

5. In Figures 2 to 4, do the metabolites (C18-C24 ceramide, CerPE, and (dh)S1P) act within the same pathway in CerS mutants? In other words, how do these alterations in sphingolipids influence each other? Is it possible that the lack of CerPE is a result of the reduced levels of C18-C24 ceramide in CerS mutants? I’d recommend double mutant analyses.

Minor:

6. Supplemental Figure 1 lacks a negative control of other tissues’ expression to confirm that CerS functions cell-autonomously for dendrite maintenance.

7. What is the potential mechanism leading to dendrite loss in CerSH215D (Fig 1E/F)? In Sup fig2, the results have excluded apoptosis, but it still have not been addressed why there is a loss in dendrite number at the LIIIw stage in CerSH215D.

8. In Figures 2 and 3, it would be beneficial to validate the results of genetic manipulations by drug inhibition or metabolite supplementation, as the authors have shown in Supplementary Figure 4 that treating C16-C18 ceramides to CerS mutant would rescue the dendritic defects. Such metabolite supplement experiment (treating with CerPE to CerS mutant), or drug inhibition (inhibiting SK1 or SK2), should be done also in figs 2-3. I put this as a minor comment provided the lipidomics be performed to supplement figures 2-4.

9. It would greatly improve the manuscript if human neuronal cell lines were also examined to verify the fly findings and the functional consequence of human CerS mutation.

10. The readability of this text could be improved. For example, the sentence "However, the fact that impairment of the function of enzymes up- or downstream of cerS leads to milder phenotypes than impairment of cerS function itself suggests that the strong phenotype in cerS mutant neurons may be due to additional metabolic problems" is too complicated. Similar issues scattered across the manuscript.

11. Does CerS activity play a role in Dscam's regulation of early developmental patterning of dendrites and axons? Similar to findings by Goyal et al., which showed that the loss of SPT in flies leads to the accumulation and mislocalization of Dscam (Down syndrome cell adhesion molecule), ultimately impairing axonal development in the adult brain [PMID 30778062]. The results in this manuscript indicate that SPT mutations also lead to impaired dendritic morphology, suggesting that the regulation of dendritic structure may be involved in a manner similar to how Dscam is regulated. Perhaps the authors could add some discussion?

Reviewer #2: Review is uploaded as an attachment

**Have all data underlying the figures and results presented in the manuscript been provided?**

Reviewer #1: Yes

Reviewer #2: Yes

PLOS authors have the option to publish the peer review history of their article (what does this mean? ). If published, this will include your full peer review and any attached files.

**Do you want your identity to be public for this peer review?** For information about this choice, including consent withdrawal, please see our Privacy Policy .

Reviewer #1: No

Reviewer #2: No

**Figure resubmission:**
---

## [Decision Letter · Decision Letter 1]

1 Aug 2025

PGENETICS-D-24-01365R1

Individual lipid alterations at the origin of neuronal Ceramide Synthase defects

PLOS Genetics

Dear Dr. Ziegler,

Thank you for submitting your manuscript to PLOS Genetics. After careful consideration, we feel that it has merit but does not fully meet PLOS Genetics's publication criteria as it currently stands. Therefore, we invite you to submit a revised version of the manuscript that addresses the points raised during the review process.

Please submit your revised manuscript within 30 days Aug 31 2025 11:59PM. If you will need more time than this to complete your revisions, please reply to this message or contact the journal office at plosgenetics@plos.org. Please include the following items when submitting your revised manuscript:

We look forward to receiving your revised manuscript.

Kind regards,

Cathy Savage-Dunn

Academic Editor

PLOS Genetics

Hua Tang

Section Editor

PLOS Genetics

Aimée Dudley

Editor-in-Chief

PLOS Genetics

Anne Goriely

Editor-in-Chief

PLOS Genetics

**Reviewers' comments:**

Reviewer's Responses to Questions

**Comments to the Authors:**

Reviewer #1: The authors have significantly improved the manuscript. I have no further questions. Recommended.

Reviewer #2: The authors have addressed all the concerns satisfactorily. It is a delight to recommend the manuscript for publication in PLoS Genetics.

Reviewer #3: The authors of this manuscript aimed to investigate the effects of disturbed lipid metabolism on the brain. To do so, they used Drosophila larvae that had specific enzymes involved in the ceramide synthesis pathway inactivated. To relate the resulting changes to Progressive Myoclonic Epilepsy Type 8 (PME8), they primarily used the morphology of Drosophila class four dendritic arborization (c4da) neurons as their metric and also used lipidomics to determine the levels of several relevant ceramide species. They found that loss of function of the Drosophila CerS enzyme not only resulted in the reduction in the levels of specific ceramide species but also neuronal degeneration. In their attempt to identify a potential therapeutic avenue, they discovered that full rescue of neuronal morphology could not only be achieved by supplementation of the missing ceramides but also requires removing the buildup of toxic CerS substrates.

Major Comments

1. Page 7, Lines 202 – 209: The order of the sub-figures does not make sense. Dendrite morphology comes first in the text but is labelled as S2F/G. The confocal images of LIII come after that in the text, but are labelled S2D/E.

2. A comparison between LIII larvae and adults is discussed on line 205, but S2D/E only shows LIII larvae. The paragraph then finishes with the missing adults, but this is labelled S2H when it should logically follow the previous sub-figure. This is confusing and the figures should be re-ordered.

3. Dorsal images of LIII larvae does not seem useful, given that these cannot be compared to the adults, which was the point being made.

Minor Comments

4. Page 3, Line 87: The lag1p “motif” is misspelled as “motive”. Please correct it

5. Page 5, Line 131 – 133: Both cited papers (10,35) discuss Hereditary Sensory Neuropathy Type I, which is most commonly caused by a mutation in the SPT1 gene, which catalyzes the initial step in sphingolipid biogenesis. While the nociceptors are likely to be affected in that disorder, it seems like a stretch to say that any genetic mutation affecting the ceramide synthesis pathway will also affect the nociceptors. This claim needs to be better supported by literature exploring other disorders of ceramide biogenesis (such as those mentioned at the start of the introduction)

6. Page 6, Line 186: The figure reference was likely meant to be “Fig. S1D” and not C. Please double check.

7. Page 6, Line 190 – 192 and Figure 1E/F: The degeneration was described as “progressive loss” but there was only a single time point studied (24 hours after the control). Without more time points in between the control and the +24 hour period, it would be impossible to say if the degeneration occurred progressively over the hours or suddenly just before the 24 hour mark. Unless you have data showing which one it might be, it might be best to simply describe it as a “loss”.

8. Supplemental Figure S1E: This figure was not cited in the text.

9. Page 7, Lines 211 – 216: Please explain why a knockdown model was used when there are mutants available that are used elsewhere in the study.

10. Page 7, Lines 213 – 214: From Figure 1G, there appears to be no significant difference between the middle group and the control. However, the text reads as though both groups were significantly different. Which is the case?

11. Figure 2D: The HexCer graph is missing its units on the Y-axis

12. Page 10, Lines 302 – 304 and Page 14, Lines 447 - 449: Given that the larvae were raised identically (presumably), would that not rule out exogenous sources? What would you say would be the likely source of difference between the two?

13. Page 10, Lines 308 – 312: In Figure S5B, it seems only C22 had a significant effect, but the text implies C18 was also significant. Which was the case? There also should be normal control to better understand how mild the improvement is in comparison to normal larvae.

14. Table S2: *** is repeated twice. The ‘<’ sign is probably missing from the last two p-values. Please double check that.

15. Page 13, Line 405: The correct figure is probably Fig 4. Please double check.

**Have all data underlying the figures and results presented in the manuscript been provided?**

Reviewer #1: Yes

Reviewer #2: Yes

Reviewer #3: Yes

PLOS authors have the option to publish the peer review history of their article (what does this mean? ). If published, this will include your full peer review and any attached files.

**Do you want your identity to be public for this peer review?** For information about this choice, including consent withdrawal, please see our Privacy Policy .

Reviewer #1: No

Reviewer #2: No

Reviewer #3: No

**Figure resubmission:**
---

## [Editor Report · Decision Letter 2]

11 Sep 2025

Dear Dr Ziegler,

We are pleased to inform you that your manuscript entitled "Individual lipid alterations at the origin of neuronal Ceramide Synthase defects" has been editorially accepted for publication in PLOS Genetics. Congratulations!

Yours sincerely,

Cathy Savage-Dunn

Academic Editor

PLOS Genetics

Hua Tang

Section Editor

PLOS Genetics

Aimée Dudley

Editor-in-Chief

PLOS Genetics

Anne Goriely

Editor-in-Chief

PLOS Genetics

Comments from the reviewers (if applicable):

**Data Deposition**

http://datadryad.org/submit?journalID=pgenetics&manu=PGENETICS-D-24-01365R2

**Press Queries**

---

## [Editor Report · Acceptance letter]

PGENETICS-D-24-01365R2

Individual lipid alterations at the origin of neuronal Ceramide Synthase defects

Dear Dr Ziegler,

We are pleased to inform you that your manuscript entitled "Individual lipid alterations at the origin of neuronal Ceramide Synthase defects" has been formally accepted for publication in PLOS Genetics! Your manuscript is now with our production department and you will be notified of the publication date in due course.

With kind regards,

Anita Estes

PLOS Genetics

On behalf of:
